


".

# Estimation of Biogenic Volatile Organic Compounds (BVOCs) Emissions in Forest Ecosystems Using Drone-Based Lidar, Photogrammetry, and Image Recognition Technologies

Xianzhong Duan[1,#], Ming Chang[1,#,*], Guotong Wu[1], Suping Situ[2], Shengjie Zhu[3], Qi Zhang[4], Yibo Huangfu[1], Weiwen Wang[1], Weihua Chen[1], and Xuemei Wang[1]

[1]Guangdong-Hongkong-Macau Joint Laboratory of Collaborative Innovation for Environmental Quality, Institute for Environmental and Climate Research, Jinan University, Guangzhou, China
[2]Foshan Ecological and Environmental Monitoring Station of Guangdong Province, Foshan, China
[3]Department of Environmental Science, Guangdong Polytechnic of Environmental Protection Engineering, Foshan, China
[4]Tianjin Academy of Eco-environmental Science, Tianjin, China
[#]These authors contributed equally to this work.

**Correspondence:** Ming Chang (changming@email.jnu.edu.cn)

**Abstract.**

Biogenic volatile organic compounds (BVOCs), as a crucial component that impacts atmospheric chemistry and ecological interactions with various organisms, play a significant role in the atmosphere-ecosystem relationship. However, traditional field observation methods are challenging to accurately estimate BVOCs emissions in forest ecosystems with high biodiversity, leading to significant uncertainty in quantifying these compounds. To address this issue, this research proposes a workflow utilizing drone-mounted lidar and photogrammetry technologies for identifying plant species to obtain accurate BVOCs emissions data. By applying this workflow to a typical subtropical forest plot, the following findings were made: The drone-mounted lidar and photogrammetry modules effectively segmented trees and acquired single wood structures and images of each tree. Image recognition technology enabled relatively accurate identification of tree species, with the highest frequency family being Euphorbiaceae. The largest cumulative isoprene emissions in the study plot were from the Myrtaceae family while monoterpenes were from the Rubiaceae family. To fully leverage the estimation results of BVOCs emissions directly from individual tree levels, it may be necessary for communities to establish more comprehensive tree species emission databases and models.

## 1 Introduction

Biogenic volatile organic compounds (BVOCs) is the medium of communication for plants to realize their wide ecological functions (Laothawornkitkul et al., 2009). BVOCs are involved in plant growth, reproduction and defense (Peñuelas and Staudt, 2010). Plants respond to the feeding of herbivores by emitting BVOCs to attract potential predators or as repellents (Kegge





and Pierik, 2010). The communication process between plants is also based on BVOCs (Šimpraga et al., 2016). For example,
gnawed plants will emit BVOCs to induce the production of defensive substances in non-attack objects (Dicke and Baldwin,
2010). In addition, BVOCs components are also used by plants to attract pollinators to bloom (Loreto et al., 2014). For the
plants themselves, under heat waves or high ozone concentrations, BVOCs seem to reduce oxidative stress and other stresses
caused by the complex non-biological urban environment (Ghirardo et al., 2016; Chen et al., 2018). At the same time, BVOCs
are emitted into the atmosphere from vegetation and have significant impacts on other organisms and atmospheric chemistry
and physics (Peñuelas and Staudt, 2010). BVOCs account for 90% of VOCs in atmospheric chemistry research which were
considered as the fuel to drive atmospheric chemical processes and the key component of the atmosphere (Heald and Kroll,
2020). The atmospheric chemical activity of BVOCs species is very sprightly, and it's lifetime usually from only a few minutes
to a few hours (Mellouki et al., 2015; Canaval et al., 2020). The contribution of BVOCs emission to global secondary organic
aerosol (SOA) generation is about 90%, which is the main source of global atmospheric SOA (Henze et al., 2008). At the same
time, BVOCs contributed about 10% $\sim$ 30% of the surface ozone in urban areas (Ran et al., 2011; Tsimpidi et al., 2012; Wu
et al., 2020; Chen et al., 2022).

However, there is considerable uncertainty (about 90% $\sim$ 120%) in the estimation of BVOCs which have important eco-
logical functions and connect the biosphere and the atmosphere (Situ et al., 2014; Wang et al., 2021). This situation re-
stricts our understanding of the atmospheric environment and ecological effects of BVOCs. Especially for the forest ecosys-
tem with the highest biodiversity, forest vegetation is considered to be the main body of BVOCs emissions, accounting for
more than 70% of global BVOCs emissions, but the uncertainty of estimation of BVOCs emissions from forest vegetation
is the most significant (Hartley et al., 2017). This uncertainty comes from two problems: the lack of field observations
and the simplification of numerical simulations. There are different methods for measuring BVOCs emissions on various
scales. At the leaf and plant scale, scholars have used confined chamber and various improved confined chamber methods
(for example, open-top chamber, free air concentration enrichment, etc) to conduct a large number of outstanding observa-
tional studies on the BVOCs emissions of leaves, branches and the whole tree and contribute different BVOCs database of
single-tree BVOCs component emissions (Isidorov et al., 1990; Komenda and Koppmann, 2002; Baghi et al., 2012; Curtis
et al., 2014). Existing potential BVOCs emission databases include seBVOCs(Steinbrecher et al., 2009), the tree BVOCs in-
dex(Simpson and McPherson, 2011), MEGAN(Guenther et al., 2012), and other general inventories (e.g. http://itreetools.org/;
http://www.es.lancs.ac.uk/cnhgroup/download.html), etc. These studies mainly quantify the emission rate of BVOCs from
specific tree species, which can help understand the processes and factors that affect the emission of BVOCs. At the forest
landscape and canopy scale, flux towers are generally established at specific forest sites to observe the BVOCs emissions of
the entire vegetation canopy (Sarkar et al., 2020). This method is relatively reliable and widely used. It can estimate vegetation
within a few hundred meters from the flux tower Canopy emission flux. Confined chamber method and flux tower observation
results can indirectly estimate the emission flux of BVOCs at an ecological scale with low biodiversity. However, for ecosys-
tems with high biodiversity such as tropical rainforest areas, this method is difficult to characterize the characteristics of such
varied vegetations.



In order to bypass the detailed investigation of ecosystem species, the academic community used aerial surveys and satellite remote sensing methods for indirect inversion of the emissions flux of BVOCs at the ecosystem and regional scales(Batista et al., 2019). However, its accuracy is relatively low, and there are still large errors. Similarly, in terms of numerical models, due to the chemical composition and species diversity of BVOCs, and are greatly affected by environmental factors, it is a big challenge to accurately simulate BVOCs emissions with numerical models. Existing numerical models (for example, BEIS, g95, MEGAN, BEM, etc.) mainly use land use, leaf biomass, emission factors, and meteorological elements to estimate BVOCs emitted by vegetation (Wang et al., 2016; Chen et al., 2022). And the key source of uncertainty in its estimation comes from the inaccuracy of the numerical model on the parameterazation and characterization of land use types, forest tree species composition, and leaf biomass. Some recent studies have found that BVOCs also have considerable spatial heterogeneity at the sub-forest scale (for example, hundreds of meters on a hillside) (Li et al., 2021). Due to the differences in the distribution of forest tree species, their BVOCs emissions are higher than those commonly assumed in biosphere emission models. More complicated. Generally, for the calculation of BVOCs emissions, accurately characterizing the spatial distribution of emission factors is a scientific difficulty that should be overcome to accurately quantitative the spatial distribution of BVOCs emissions.

In recent years, consumer-grade UAV platforms, lidar measurement technology and computer image recognition technology have developed rapidly. UAVs equipped with measuring instruments for rapid sample observation technology gradually mature, and its positioning accuracy can reach the centimeter level. Even in areas such as forest protection areas, it is possible to set up routes to carry out surveys based on suitable forest gaps. UAVs equipped with sensors to measure atmospheric components have also begun to emerge (Villa et al., 2016). Many scholars install sensors in drone-based platforms for low-cost and flexible measurement of VOC, black carbon (BC), ozone, aerosol particles, etc (Brosy et al., 2017; Rüdiger et al., 2018; Shakhatreh et al., 2019; Li et al., 2021; Wu et al., 2021). And the camera carried by the drone can also obtain very high-resolution images, and even multi-spectral images (Nebiker et al., 2008; Villa et al., 2016; Dash et al., 2017). At the same time, the miniaturization of lidar measurement technology also makes it possible to be carried by UAVs (Zhao et al., 2016). Lidar as the instrument with the highest surveying accuracy so far. Compared with the original measurement method, the point cloud obtained by lidar can characterize the canopy structure of each tree in the measurement range in a revolutionary way (Li et al., 2012; Jin et al., 2021). The characterization of forest community structure, morphological and physiological forest traits has been greatly enriched by the combined laser scanning and imaging spectroscopy (Schneider et al., 2017).

With the rapid development of computer image recognition technology, the recognition of plant species is based on the method of machine vision recognition of plants in plant phenomics to directly characterize plant species (Fassnacht et al., 2016; Cheng et al., 2023). Usually, machine learning and deep learning methods are used to call plant image libraries to train machine vision interpretation learning models, and then violently interpret high-resolution multi-spectral remote sensing images and laser point clouds to obtain accurate plant populations and species result (Sylvain et al., 2019). At present, there are several vegetation species classifiers have been applied: logistic regression, linear discriminat analysis, random forest, support vector machines, k-nearest neighbors (kNN), and 2d or 3d convolutional neural networks (CNNs) (Michałowska and Rapiński, 2021). With the maturity of various technologies and recognition training databases, various communities have created a batch of open source, shared, and API-callable recognition apps or platform for the public. The users only need to upload





photos to get the recognized result, and the accuracy is quite good. Open source recognition tools for lidar results have also been developed rapidly. The accuracy of species classification methods based on structural features based on LiDAR height, intensity, and combination of height and intensity parameters can reach from 87% to 92% (You et al., 2020). Many publications have proven that the combination of LiDAR data and multispectral or hyperspectral images produces higher accuracy of species classification compared to LiDAR data alone (Michałowska and Rapiński, 2021).

Therefore, this research intends to establish a technical framework based on the lidar and photogrammetry carried by drones and image recognition technologies from community to identify plant species to obtain accurate BVOCs emissions. It is expect that the combination of the Lidar accurate characterization technology of forest canopy, the ascendant accurate identification technology of tree species, and the tree-species emission factor database obtained from long-term surveys, could creates a new way to accurately quantify the biogenic emissions.

## 2 Methods

### 2.1 The Description of Workflow

The entire workflow includes the following aspects (as shown in Fig.1) : The first is the selection of drones equipped with lidar and high-resolution cameras; the second is the interpretation of photogrammetry results. The third part is to give the images of each tree to API-callable plant species identify platforms, and then establish a match between the interpreted tree species and the single tree species BVOCs emission factor database; the fourth step is to calculate the BVOCs emissions of the study area based on the match results and emission factors.

### 2.2 Study Area

The location of the provision of the work is in the coniferous and broadleaved mixed forest of the Dinghushan Forest Ecosystem Research Station of Chinese Ecosystem Research Network (CERN). Dinghu Station is located in South Subtropical Zone, belongs to Subtropical Tropical quarter wind moisturized climate, and winter and summer climate is obvious. The average annual temperature is 20.9 °C, the average annual rainfall is 1900 mm, and the annual sun radiation is about 4665 MJ·m$^{-2}$·year$^{-1}$, and the average annual sunshine time is 1433 hours, and the average annual evaporation amount is 1115 mm, and the average relative humidity in many years is 82%. The position is near the northern return line, and its elevation is 300∼350 meters while the slope is about 25°∼30°, and the slope direction is south. Its soil is Lateritic red soil, the soil layer depth is about 30 cm to 90 cm. The plots have a long-term appearance of the tree species, which is easy to compare the test results. There are 260 families, 864 genus, 1740 species, and 349 species of cultivated plants in Dinghushan Forest. At the same time, Li et al. (2021) used drones equipped with online mass spectrometers at Dinghushan Station to observe the composition of VOCs. We hope to compare their results to explore the influence of tree species on the spatial heterogeneity of VOCs.





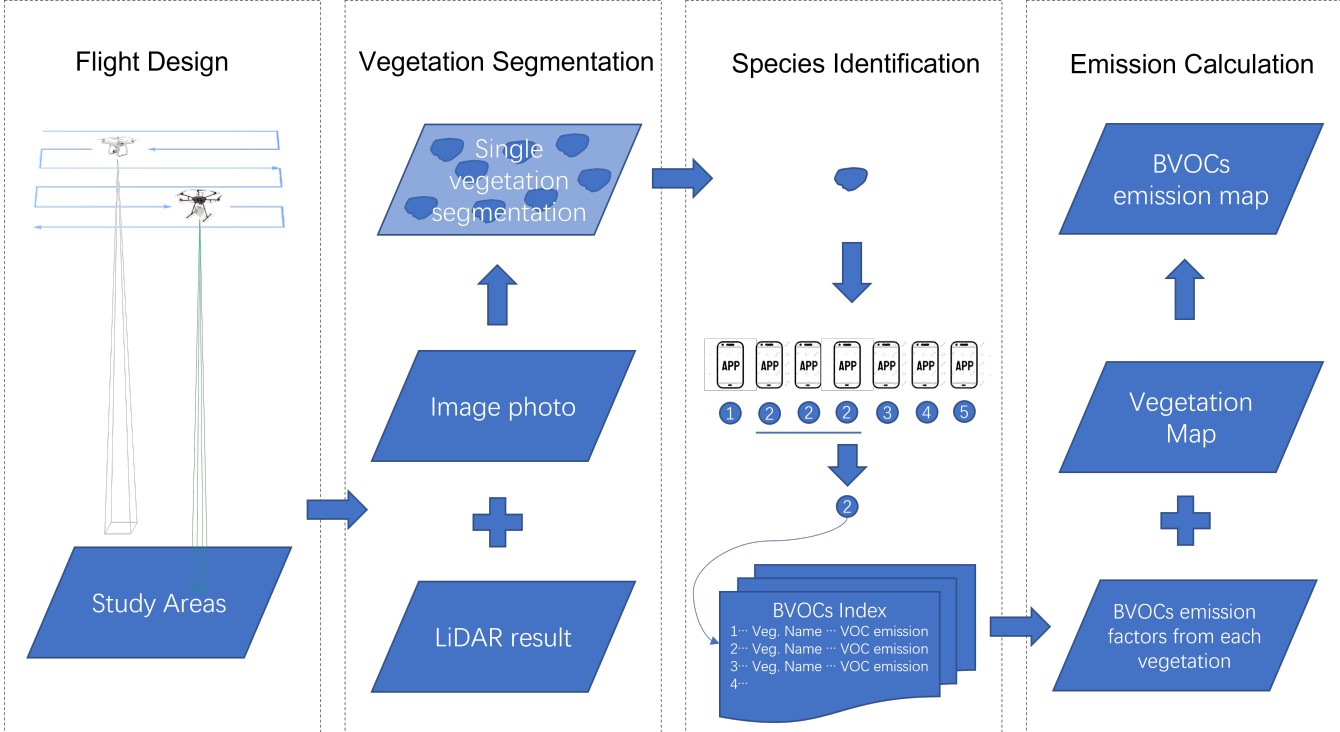

**Figure 1.** Schematic workflow of this study

## 2.3 Flight Equipment and Instruments

The main UAV platform used in this technical framework is DJI® Matrice 600 Pro, which is an universal platform that can carry various sensors. We equipped the GreenValley® LiAir V lidar scanning system on this platform, which also includes a set of integrated navigation system composed of global navigation satellite systems(GNSS), inertial measurement unit(IMU), and attitude calculation software.

At the same time, we simultaneously used a DJI® Phantom 3 Professional UAV to get visible light images. Its camera model is FC300X_3.6_4000×3000(RGB) , and the camera image sensor (CMOS) is $1/2.3$ inch which effective pixels is 12.4 million (total pixels 12.76 million). According to the image attribute information, the camera parameters used in this work are: aperture value $f/2.8$, maximum aperture 2, exposure time $1/1250$ second, ISO speed 100, focal length 4 mm.

The DJI® pilot software are employed to design the flight route and guide the flight of the UAVs. At the beginning of the take-off phase, we use manual operation to ensure that the trees near the forest gap are avoided, and after the takeoff reaches the specified height, it changes to automatic flight (as shown in Fig. 2). The flight mode of the two planes is designed as a same flight route, so that they can obtain a consistent measurement area.



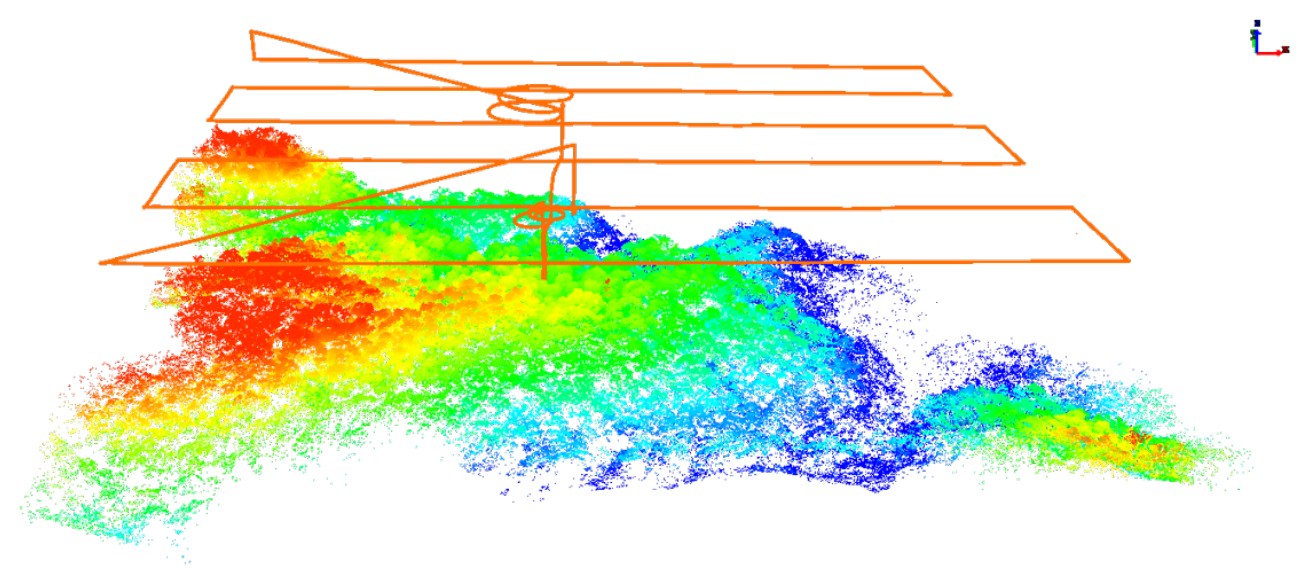

**Figure 2.** Flight route of this study

## 2.4 LiDAR-Based Tree Segmentation and Canopy Structure Calculation

The study specifically uses GreenValley® LiDAR360 and Esri® ArcGIS software to carry out its work. First, the laser point cloud results are coordinated and spliced, and then the noise is removed when it overload 5 times the standard deviation, and then the improved progressive TIN densification (IPTD) algorithm is used to separate the ground points (Zhao et al., 2016). On this basis, a digital elevation model is generated based on the inverse distance weight (IDW) method (Ismail et al., 2016).

The processing of obtaining single tree features based on lidar is based on the layer stacking algorithm (Ayrey et al., 2017). According to the layer height of different trees, the position of the seed point in the laser point cloud is determined for segmentation, and then the boundary of each tree is obtained. Then, the characteristic parameters of tree canopy structure, such as canopy height, canopy radius, etc., are calculated. Based on the individual tree boundary, the results of the visible light image segmentation through the overlay analysis of ArcGIS are used to obtain the image picture of each individual tree. After that, the pictures of individual trees are given to different APPs to obtain the plant species identification results.

## 2.5 Vegetation Identification

With the continuous improvement of a new generation of plant recognition algorithms based on deep learning methods, a variety of plant recognition APPs and platforms continue to appear (Irimia et al., 2020; Otter et al., 2021). They can all import and identify plant images in the mobile phones or give a application programming interface (API) to the public researchers.



**Table 1.** List of plant species identification apps and platforms

| Name | Source | Reference |
| --- | --- | --- |
| AiPlants | http://hbl.nongbangzhu.cn/ | Zhanhui et al. (2020) |
| Aliyun GIRS | https://vision.aliyun.com/ | Jin (2017) |
| Baidu EasyDL (PaddlePaddle) | https://cloud.baidu.com/ | Ma et al. (2019) |
| LeafSnap | http://leafsnap.com/ | Kumar et al. (2012) |
| Pl@ntNet | https://identify.plantnet.org/ | Joly et al. (2016) |
| PlantSnap | https://www.plantsnap.com/ | Otter et al. (2021) |
| Tree-detection-evo | https://github.com/jaeeolma/tree-detection-evo/ | Mäyrä et al. (2021) |

There are also quite a lot of open source deep learning trained models and datasets, allowing researchers to submit visible light images and obtain recognition results (Ma et al., 2019; Zhanhui et al., 2020).

The apps and platforms shown in the table 1 were used in this study to identify the visible light image after point cloud segmentation. They are usually trained based on a certain national or international plant classification picture database. For example, the AiPlants[®] is based on the database of Plant Photo Bank of China (PPBC) (Zhanhui et al., 2020). With the rise

of cloud computing services, there have provided their own calling methods on various platforms, such as Aliyun[®] general image recognition service (GIRS), Amazon[®] rekognition service, Baidu[®] paddle-paddle platform, etc. And their identification results can be obtained using simple script submission (Jin, 2017). However, whether the reliability, accuracy, and portability of these simple retrieval methods can support their application in the survey of plant emissions needs more explore.

### 2.6 BVOCs Emission Factor and Emission Calculation

In this study, we calculated based on the database of detailed BVOCs emission factors (EF) for tree species provided by MEGAN3.2 which contains a set of EF libraries with more than 40,000 tree species (Guenther et al., 2018). When the tree species determined based on section 2.5 is clear, the corresponding BVOCs EF can be obtained by looking up the table.

For the types of trees that are not contained in the EF library, we obtain the BVOCs emission factor of the tree species based on the literature survey method (Chen et al., 2022; Mu et al., 2022). For tree species that still cannot be found even through

literature surveys, we choose plants of the same *Gemera* to replace.

Because there are quite a few types of BVOCs obtained by observation experiments, they are generally dominated by isoprene and monoterpenes (Li et al., 2021). Therefore, our study is also characterized by the distribution of emissions using its genus-specific average emission factor.

Since the images we use to identify tree species are single temporal, we only attempt to calculate the maximum and minimum

emissions of the forest in the sample plot. The calculation method is based on the emission factors corresponding to the species of each tree, multiplied by its biomass and the area occupied by its crown diameter.



# 3 Results

## 3.1 The morphological composition of the vegetation

Based on the point cloud results measured by lidar, more accurate arbor morphological characteristics can be obtained. Then
we split the individual trees, and the point clouds of each individual tree are shown in Fig. 3. It can be seen that due to the influence of terrain, the point cloud at the edge has a much lower density than the point cloud in the center, which may cause higher uncertainty in the segmentation of the single-wood in this area.

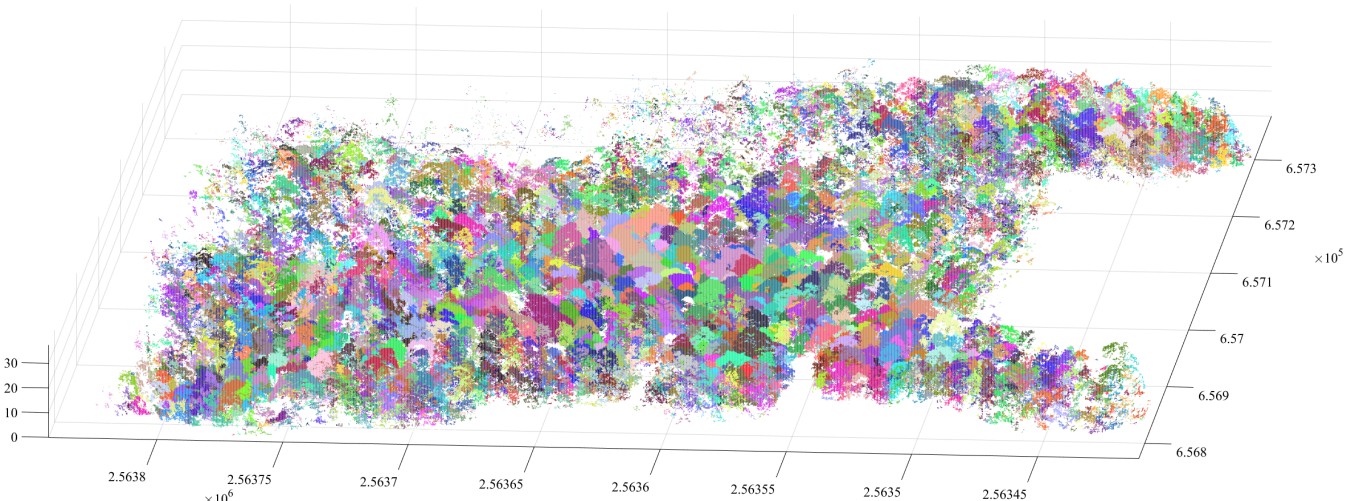

**Figure 3.** Point cloud of each individual tree obtained based on layer stacking algorithm excluding topographic

After the statistics of single tree segmentation, there are 1291 trees in the sample plot. The overall distribution of morphological parameters and the corresponding relationship between tree height and crown diameter of each tree are shown in Fig.4.
It can be seen that the tree height in the sample plot obtained by the measurement follows the GaussAmp skew distribution. Its distribution range spans from 2 m to 30 m, and its average is at 14.9 m. At the same time, its crown radius presents a lognormal distribution, and its average value is about 4 m.

## 3.2 The composition of vegetation species

The plant identification APPs were called to identify the tree species based on the segmentation results. The spatial distribution
and frequency of tree species are shown in Fig.5 and Table 2 It can be seen from its spatial distribution that different tree species appear to be scattered and gathered. Among them, the top three frequency tree species is *Aidia canthioides (Champ. ex Benth.) Masam.*, followed by *Macaranga sampsonii Hance*, and third is *Blastus cochinchinensis Lour.* while the highest frequency family is *Euphorbiaceae*. The ratio of top three species is about 12%, 11% and 6%. Other identified tree species are also shown in Table 2. Combined with their canopy morphology distribution, it can be seen that the plot presents significant





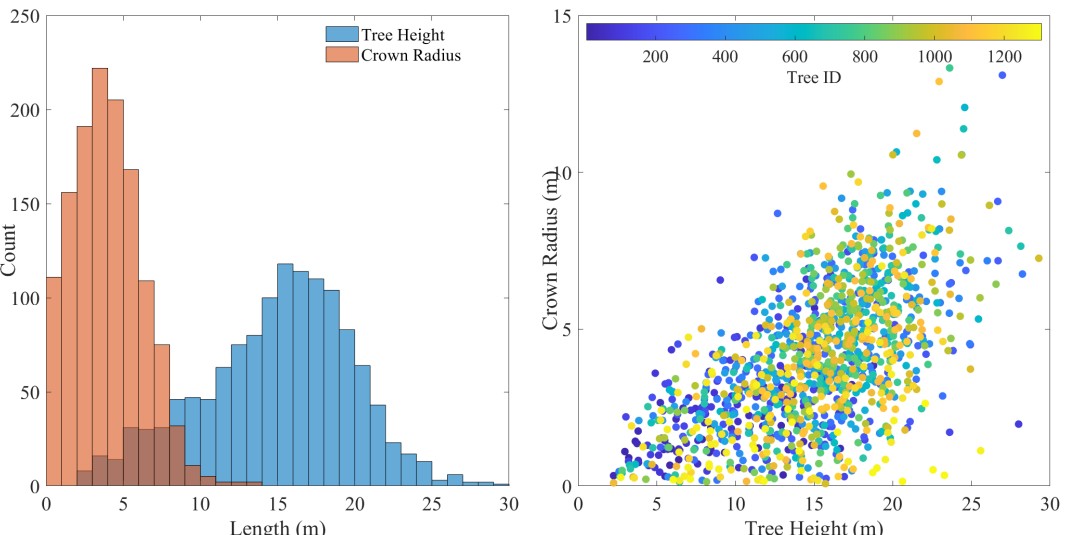

**Figure 4.** The distribution of tree height and crown radius (left: overall distribution; right: each tree)

coniferous and broad-leaved mixed forest characteristics, and coniferous/broad-leaved trees occupy the position of dominant tree species. Meanwhile, it still can be see from Fig.5 that lots of trees could not recognized.

### 3.3 The BVOCs emission in family and individualized scale

The emission we obtained for isoprene and monoterpene for each family are shown in Table 3. It can be seen that in the study area of Dinghu Mountain, the largest cumulative isoprene emissions were from *Myrtaceae* family (maximum 18.7

$\mu g C m^{-2} h^{-1}$), followed by *Salicaceae* family (maximum 3.8 $\mu g C m^{-2} h^{-1}$), while for monoterpenes their cumulative emissions were largest in *Rubiaceae* family (maximum 3.9 $\mu g C m^{-2} h^{-1}$), followed by *Theaceae* family (maximum 2.8 $\mu g C m^{-2} h^{-1}$). However, it is worth noting that since we cannot confirm the leaf type, leaf age, and corresponding phenological period of each tree, we only calculated the maximum and minimum possible emissions based on their standard emission factors and biomass.

At the same time, the spatial distribution of individual plant emissions from Fig.6 shows that there are clusters of BVOCs

emitting plants in the study area, which are caused by the aggregation of plants of the same family. The clusters of isoprene and terpene emitting plants are homogeneous, while there are some non-BVOCs emitting plants between the different clusters, which may be related to their ecological competition strategy (Fitzky et al., 2019). According to the forest competition theory, the emission of BVOCs is related to its competitive pressure, relative size and area overlap rate (Contreras et al., 2011). On the other hand, the strategies adopted by different species are different. The intra-specific competition and inter-specific

competition play a specific role through different biopheromones which are all BVOCs (Šimpraga et al., 2019). In addition, it is noteworthy from Fig.6 that the number of plants not discriminated in the study area is quite large, implying that this is an important source of uncertainty in the estimation of BVOCs emissions in this method.







**Figure 5.** The spatial distribution of tree species

# 4 Discussion

## 4.1 The uncertainties sources of this method

### 4.1.1 Flight altitude and image resolution

During the field flight using this workflow, we found that the height of the flight and the pixel area occupied by each tree in the resulting visible light image is the decisive link that determines whether the image recognition tool can effectively identify the plant species in the image . In practice flight, we designed different flight altitude routes, namely 60 meters, 120 meters and 200 meters, in order to find a suitable flight altitude. We checked and found that for the images obtained at a flying altitude of 120 meters or more, the number of pixels per tree obtained after being cut and paired by a single tree in the lidar point cloud is less (about 200*300 pixels). The description of tree leaf characteristics is very unclear and presents mosaic-like characteristics,







**Figure 6.** The individualized spatial distribution of isoprene and monoterpenes emission factor

which makes it impossible to accurately identify the hidden plant species in different image recognition tools, which also makes the subsequent calculation of BVOCs emissions into trouble. Especially since the images obtained from the aerial survey of





drones are all looking down, the expression of canopy morphological features is lacking. This requires further integration with
lidar results, or the use of multi-spectral and multi-angles cameras in the future.

### 4.1.2    The selection of image recognition tool

It can be seen that the recognition accuracy of these APPs is not as high as it claims for the visible light images obtained
by drones. Among them, platforms trained by satellite images give quite accurate results of tree specie recognition. EasyDL
gave back "unrecognizable" feedback for quite a lot of individual tree images while AiPlants and LeafSnap gave incorrect
classification and recognition results such as succulents, garden plants, etc. This result may related to the fact that the APPs did
not train image inputs with right tree species tags during the collection and training of the general image datasets. In general,
for the refined calculation of vegetation VOCs emission factors, the platform based on deep learning training from remote
sensing images can provide faster and more reliable tree species identification results than traditional methods.

### 4.1.3    The BVOCs emission factor database

The emission factors of various tree species used in this study mainly come from Mu et al. (2022) measured results database.
Although this literature has rarely measured over a hundred tree species in South China, there are still considerable shortcom-
ings. Firstly, there are still quite a few parsed tree species that are not within the scope of this database in this study; Secondly,
the emissions of BVOCs from trees are subject to various photochemical and hydrothermal conditions, but currently, various
databases are unable to provide detailed characterization of the impact of these environmental factors at the tree species level;
Thirdly, different BVOCs emission factor databases have different emphasis on the emission parameters of BVOCs compo-
nents of the same tree species. These deficiencies limit our further application and migration of this method to other forests.
Community peers can only refer to our Technology roadmap and their local tree species emission factor library for further
estimation of BVOCs emissions.

### 4.2    The differences of BVOCs emission and its potential impact

The BVOCs obtained in this study were compared with the emission and concentration results obtained by different methods
at the sample site of Dinghu Mountain, as shown in the Table 4. It is interesting to see that the ways of the different studies
are very diverse (including model calculations, forest floor sampling observations, drone-mounted sensor observations, etc.).
The quantitative differences in the emissions of BVOCs from Dinghu Mountain obtained by the methods of the different
studies are significant. The results obtained in this study show that a more BVOCs emissions of forest biodiversity allows the
calculated results to be more consistent with those obtained from direct observations carried out in the forest canopy. Previous
estimates of biomass based on a simple PFT approach may have underestimated BVOCs emissions to a considerable extent.
This places a new requirement on the regional-scale estimation of BVOCs emissions to consider the forest's biodiversity in the
region when modeling regional-scale BVOCs emissions and to consider vegetation factors beyond purely physical canopy size
representations such as LAI and crown diameter.



## 5 Conclusions

This research has established a workflow for identifying plant species based on lidar, photogrammetry and image recognition technologies carried by drones to obtain accurate BVOCs emissions. The innovation of this research is to combine the newly developed rapid survey method of plant species with the calculation of BVOCs emissions, and discussed the main uncertainty sources of the BVOCs emissions obtained in this method. The current limitation of this study is that although LiDAR can capture the multi-layer structure of tree crowns, visible light is difficult to identify other vegetation below trees, such as shrubs and herbs, which can result in a certain loss of BVOCs emissions.

The inspiration of this study is that with the development of new technologies in computer science, tree species identification, which previously restricted the estimation of BVOCs emissions, will gradually be able to achieve breakthroughs through large-scale image recognition technology. However, at the same time, the open source and standardized image recognition technology methods, as well as the BVOCs emission factor library of tree species, have become new bottlenecks, and relevant research communities need to consider how to share corresponding data and technologies more openly.

*Author contributions.* DX: Data curation, Writing – original draft preparation; MC: Supervision, Conceptualization, Methodology, Visualization, Writing – original draft preparation; GW: Data curation, Software, Validation, Visualization; SS: Formal analysis, Investigation; SZ: Resources, Drone Techniques; QZ: Data curation; YH:Writing – review & editing; WW: Writing – review & editing; WC: Writing – review & editing; XW: Project administration

*Competing interests.* The authors declare that they have no conflict of interest.

*Acknowledgements.* This work was supported by the National Key Research and Development Program of China (2023YFC3706202), the National Natural Science Foundation (42275107, 41905086, 41705123), the Science and Technology Projects in Guangzhou (2023A04J0251), the Special Fund Project for Science and Technology Innovation Strategy of Guangdong Province (Grant No.2019B121205004). The authors thank the Dinghushan Forest Ecosystem Research Station for provide sample plots.



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



**Table 2.** Specific species composition information of vegetation identification result

| Families | Gemera | Species | Count | Mean Height (m) | Mean Crown Radius (m) |
|---|---|---|---|---|---|
| Actinidiaceae | Saurauia | Saurauia tristyla DC. | 3 | 7.4 | 0.9 |
| Aquifoliaceae | Ilex | Ilex cochinchinensis (Lour.) Loes. | 1 | 17.2 | 6.7 |
| Araliaceae | Schefflera | Schefflera heptaphylla (Linnaeus) Frodin | 5 | 10.1 | 2.9 |
| Arecaceae | Caryota | Caryota maxima Blume ex Martius | 2 | 13.7 | 2.7 |
| Burseraceae | Canarium | Canarium album (Lour.) Rauesch. | 6 | 19.8 | 4.5 |
| Cannabaceae | Gironniera | Gironniera subaequalis Planch. | 19 | 18.0 | 5.6 |
| Celastraceae | Euonymus | Euonymus laxiflorus Champ. ex Benth. | 2 | 15.1 | 3.7 |
| Ebenaceae | Diospyros | Diospyros eriantha Champ. ex Benth. | 3 | 10.9 | 2.1 |
| Ericaceae | Craibiodendron | Craibiodendron scleranthum (Dop) Judd. | 2 | 10.6 | 1.6 |
| Euphorbiaceae | Macaranga | Macaranga sampsonii Hance | 67 | 12.5 | 1.9 |
| | | Macaranga andamanica Kurz | 4 | 9.4 | 0.3 |
| | Mallotus | Mallotus paniculatus (Lam.) Muell. Arg. | 22 | 12.3 | 2.7 |
| Fabaceae | Ormosia | Ormosia glaberrima Y. C. Wu | 20 | 17.8 | 5.4 |
| | Archidendron | Archidendron lucidum (Benth) I. C. Nielsen | 6 | 18.2 | 7.6 |
| Fagaceae | Castanopsis | Castanopsis chinensis (Sprengel) Hance | 6 | 15.9 | 5.0 |
| Juglandaceae | Engelhardtia | Engelhardia roxburghiana Wall. | 3 | 11.1 | 2.8 |
| Lauraceae | Cryptocarya | Cryptocarya concinna Hance | 17 | 18.2 | 7.2 |
| | | Cryptocarya chinensis (Hance) Hemsl. | 9 | 11.2 | 0.8 |
| | Lindera | Lindera chunii Merr. | 5 | 17.7 | 5.8 |
| | Machilus | Machilus chinensis (Champ. ex Benth.) Hemsl. | 3 | 12.6 | 3.6 |
| | Neolitsea | Neolitsea cambodiana Lec. | 1 | 15.1 | 5.5 |
| Malvaceae | Pterospermum | Pterospermum lanceifolium Roxburgh | 19 | 12.0 | 2.4 |
| | | Pterospermum heterophyllum Hance | 4 | 15.2 | 3.6 |
| Melastomataceae | Blastus | Blastus cochinchinensis Lour. | 35 | 18.9 | 6.8 |
| | Memecylon | Memecylon ligustrifolium Champ. | 2 | 6.2 | 1.9 |
| Moraceae | Ficus | Ficus esquiroliana Levl. | 9 | 18.1 | 5.7 |
| | | Ficus nervosa Heyne ex Roth | 2 | 5.6 | 1.8 |
| Myrtaceae | Syzygium | Syzygium rehderianum Merr. et Perry | 29 | 15.6 | 3.7 |
| | | Syzygium acuminatissimum (Blume) Candolle | 10 | 15.3 | 5.5 |
| | | SyzygiumlevineiMerr. et Perry | 2 | 9.3 | 2.8 |
| | | Syzygium championii (Benth.) Merr. et Perry | 1 | 18.1 | 4.8 |
| Pandaceae | Microdesmis | Microdesmis caseariifolia Planch. | 6 | 14.0 | 3.2 |
| Phyllanthaceae | Aporosa | Aporosa yunnanensis (Pax & K. Hoffmann) F. P. Metcalf | 34 | 13.2 | 2.8 |
| | Bridelia | Bridelia balansae Tutcher | 4 | 13.2 | 3.8 |
| Polygalaceae | Xanthophyllum | Xanthophyllum hainanense Hu | 15 | 16.9 | 3.8 |
| Primulaceae | Ardisia | Ardisia quinquegona Bl. | 13 | 16.2 | 4.1 |
| | | Ardisia waitakii C. M. Hu | 3 | 15.2 | 6.3 |
| Rhizophoraceae | Carallia | Carallia brachiata (Lour.) Merr. | 2 | 15.4 | 3.1 |
| Rosaceae | Pygeum | Pygeum topengii Merr. | 7 | 13.2 | 4.9 |
| Rubiaceae | Aidia | Aidia canthioides (Champ. ex Benth.) Masam. | 68 | 17.1 | 5.6 |
| | Lasianthus | Lasianthus chinensis (Champ.) Benth. | 6 | 13.0 | 2.8 |
| | Peponidium | Canthium horridum Bl. Bijdr. | 5 | 11.7 | 1.0 |
| | Psychotria | Psychotria rubra (Lour.) Poir. | 5 | 14.1 | 2.1 |
| | Canthium | Canthium dicoccum(Gaertn.) Teysmann et Binnedijk | 3 | 12.6 | 2.7 |
| Rutaceae | Acronychia | Acronychia pedunculata (L.) Miq. | 2 | 17.2 | 4.0 |
| Sabiaceae | Meliosma | Meliosma rigida Sieb. et Zucc. | 3 | 9.2 | 2.8 |
| Salicaceae | Casearia | Casearia glomerata Roxb. | 2 | 13.9 | 0.8 |
| Sapindaceae | Mischocarpus | Mischocarpus pentapetalus (Roxb.) Radlk | 24 | 11.8 | 1.2 |
| Sapotaceae | Sarcosperma | Sarcosperma laurinum (Benth.) Hook. f. | 8 | 16.9 | 3.7 |
| Theaceae | Schima | Schima superba Gardn. et Champ. | 3 | 7.8 | 0.7 |
| Thymelaeaceae | Aquilaria | Aquilaria sinensis (Lour.) Spreng. | 2 | 17.2 | 7.8 |



**Table 3.** The maximum and minimum emissions from different families in study area (Unit: $\mu g C m^{-2} h^{-1}$))

| Families | Isoprene | | Monoterpenes | | Count of trees |
|---|---|---|---|---|---|
| | Minimum | Maximum | Minimum | Maximum | |
| Actinidiaceae | 0.0 | 0.0 | 0.0 | 0.0 | 3 |
| Aquifoliaceae | 0.0 | 0.0 | 0.0 | 0.0 | 1 |
| Araliaceae | 0.0 | 0.0 | 0.0 | 0.0 | 5 |
| Arecaceae | 1.6 | 1.6 | 0.0 | 0.0 | 2 |
| Burseraceae | 0.1 | 2.9 | 0.0 | 0.0 | 6 |
| Cannabaceae | 0.0 | 0.0 | 0.0 | 0.0 | 19 |
| Celastraceae | 0.0 | 0.0 | 0.0 | 0.0 | 2 |
| Ebenaceae | 0.0 | 0.0 | 0.0 | 0.1 | 3 |
| Ericaceae | 0.0 | 0.0 | 0.0 | 0.0 | 2 |
| Euphorbiaceae | 0.6 | 0.9 | 0.1 | 0.1 | 93 |
| Fabaceae | 0.8 | 0.8 | 0.2 | 0.2 | 26 |
| Fagaceae | 0.0 | 0.0 | 0.0 | 0.0 | 6 |
| Juglandaceae | 0.0 | 0.0 | 0.0 | 0.0 | 3 |
| Lauraceae | 2.2 | 2.8 | 2.4 | 2.6 | 35 |
| Malvaceae | 0.0 | 0.0 | 0.0 | 0.0 | 23 |
| Melastomataceae | 0.0 | 0.0 | 0.0 | 0.0 | 37 |
| Moraceae | 0.1 | 0.4 | 0.0 | 0.0 | 11 |
| Myrtaceae | 0.7 | 18.7 | 0.0 | 0.8 | 42 |
| Pandaceae | 0.0 | 0.0 | 0.0 | 0.0 | 6 |
| Phyllanthaceae | 0.0 | 0.0 | 0.0 | 0.5 | 38 |
| Polygalaceae | 0.0 | 0.0 | 0.0 | 0.0 | 15 |
| Primulaceae | 0.0 | 0.0 | 0.0 | 0.1 | 16 |
| Rhizophoraceae | 0.0 | 0.0 | 0.0 | 0.0 | 2 |
| Rosaceae | 0.0 | 0.0 | 0.0 | 0.0 | 7 |
| Rubiaceae | 0.0 | 0.3 | 0.0 | 3.9 | 87 |
| Rutaceae | 0.0 | 0.0 | 0.0 | 0.1 | 2 |
| Sabiaceae | 0.0 | 0.0 | 0.0 | 0.0 | 3 |
| Salicaceae | 0.7 | 3.8 | 0.0 | 0.0 | 2 |
| Sapindaceae | 0.0 | 2.9 | 0.0 | 0.7 | 24 |
| Sapotaceae | 0.0 | 1.8 | 0.0 | 0.0 | 8 |
| Theaceae | 0.0 | 0.0 | 2.8 | 2.8 | 3 |
| Thymelaeaceae | 0.0 | 0.0 | 0.0 | 0.0 | 2 |
| **Total** | **7.0** | **37.1** | **5.8** | **12.1** | **534** |



**Table 4.** Measurements and simulations of isoprene and monoterpenes emissions/concentrations using different methods at the same site

| Methods | Isoprene | Monoterpene | Reference |
|---|---|---|---|
| This study | $7.0 \sim 37.1 \, \mu g C m^{-2} h^{-1}$ | $5.8 \sim 12.1 \, \mu g C m^{-2} h^{-1}$ | - |
| MEGAN | $0.1 \sim 10 \, \mu g C m^{-2} h^{-1}$ | $0.1 \sim 10 \, \mu g C m^{-2} h^{-1}$ | (Guenther et al., 2012) |
| REA techniques | $0.11 \, mg C m^{-2} h^{-1}$ | $0.24 \, mg C m^{-2} h^{-1}$ | (Gao et al., 2011) |
| REA techniques | $0.215 \, mg C m^{-2} h^{-1}$ | $0.313 \, mg C m^{-2} h^{-1}$ | (Situ et al., 2013) |
| GC-MS | $0.12 \pm 0.80$ ppbv | $0.32 \pm 0.16$ ppbv ($\alpha$-pinene) | (Tang et al., 2007) |
| GC-MS | $0.76 \pm 0.50$ ppbv | $0.33 \pm 0.18$ ppbv ($\alpha$-pinene) | (Wu et al., 2016) |
| UAV-based VOC sampler | $0.047 \pm 0.040$ ppbv | $0.084 \pm 0.104$ ppbv ($\alpha$-pinene) | (Li et al., 2021) |