# Peer review of "Estimation of Biogenic Volatile Organic Compounds (BVOCs) Emissions in Forest Ecosystems Using Drone-Based LiDAR, Photogrammetry, and Image Recognition Technologies"

_Atmospheric Measurement Techniques, 2024_

## Author Comment (AC1)

**Response to Referee**

RC1: 'Comment on amt-2024-25', Anonymous Referee #1, 26 Mar 2024

This study proposed and established a technical framework based on the lidar and photogrammetry carried by drones, utilizing image recognition technologies to identify plant species to obtain accurate BVOCs emissions. It is expected that the combination of the Lidar characterization technology, the identification technology of tree species, and the tree-species emission factor database could create a new way to accurately quantify the biogenic emissions over a large region. However, in current form, details of technique and the uncertainty discussion are somewhat less satisfactory for AMT journal. In addition, the language of this manuscript need refinement. Overall, I suggest providing more information on method descriptions and uncertainty sources before the manuscript can be accepted. Specific suggestions are listed below.

Thank you for the reviewer's comments and suggestions. We have made significant revisions to the paper according to the reviewer's comments. Specific point-to-point modifications, such as subsequent blue font text and corresponding difference files.

Line 127: What does the 'forest gap' mean? Why is it needed to avoid the gap?

Thank you for the comments. This is the problem caused by our unclear expression when translating into English. The original meaning here is as follows: the airborne lidar we use needs to first complete inertial guidance to ensure surveying accuracy, and usually needs to fly out of a circular shape during takeoff. However, on the underlying surface of the forest, there are various restrictions on the landing site of drones, the most prominent of which is that the forest canopy is too dense, which poses a risk of crashing during the landing process. At this point, it is necessary to find a "forest gap" to reduce this risk. And the forest gap is generally choosing as a tomb, ridge, or other natural bare ground. This situation also restricts the application of this method in forests in different regions.
We revised this part as follow:
"It is worth noting that in forest areas, due to the dense layers of trees, there are significant risks during takeoff and landing, so it is usually necessary to find a suitable landing location. We usually choose the location at the "forest gap", which is usually a tomb, ridge, or other natural bare ground. At the beginning of the takeoff phase, we used manual operation to avoid trees near the forest gap to reduce the risk of a crash, while completing inertial guidance for the IMU. After the takeoff reaches the specified height, it changes to automatic flight."

Method 2.4: Since the drone would obtain large number of photos on different terrain and elevations, it is important to reconstruct the whole targeted area from each photo and avoid the replicated identification of every single tree. Please provide more details on how to combine the information of individual tree location from lidar data with

photo taken.

Thank you for your suggestion. We also believe that this is important to reconstruct the whole targeted area from each photo and avoid the replicated identification of every single tree in the process of identifying. Therefore, we adopted a relatively mature airborne LiDAR method to first obtain a large number of laser point clouds, and then perform single tree segmentation based on layer stacking algorithm. The principle of this algorithm is to first obtain the seed points of each single tree and then find its watershed (Li et al., 2012). This single tree segmentation technique has been widely used in various forestry projects based on airborne LiDAR. At the same time, we fuse and concatenate the airborne visible light image into a complete image raster data, and then segment it based on the boundary layer of single tree segmentation. The specific parameter settings for our airborne image data processing are shown in the table below. We have also supplemented the paper according to your suggestions.

Table 1 The specific parameter settings for airborne image data processing

| Parameter | Value | Unit |
| --- | --- | --- |
| Ground sample distance | 7.2 | cm |
| Overlap in flight direction | 85% | - |
| Side overlap | 60% | - |
| Aera Coverd | 0.372 | $km^2$ |
| Mean absolute geolocation variance | 0.0138-0.0361 | cm |
| Mean point density | 42.6 | $point/m^2$ |

Methods: Although this study provide an innovative method to recognize BVOC emission from drone-based lidar and camera, details of specific techniques used here still require further clarification, for example, how to design the flight route, the lidar data processing and the algorithm of identifying tree species.

Thank you for the reviewer's suggestions. We fully agree with the reviewer's view that the focus of this study is to provide a new methodological framework for estimating BVOCs emissions, which integrates multiple relatively mature methods from different scientific fields. Therefore, the details of each technology were not previously presented. For example, designing automatic drone routes based on the scope of the research area is already built-in in DJI's software, and corresponding professional software has been developed for processing airborne lidar and identifying tree species, respectively. Thus, we provide additional setup details and discussion of them in each related section of route design, lidar data processing, and methods for identifying tree species.

Discussion 4.1: The authors presented several types of uncertainty sources. Then it is curious for readers to know the dominant uncertainty and the exact uncertainty level. In addition, authors could propose some possible solution and research directions to mitigate these uncertainties on emission estimation.

Thank you for the reviewer's suggestions. We also believe that the lack of direct quantification of uncertainty levels for each source is a major issue in this paper. We believe that this is caused by two factors. On the one hand, the entire technical framework process is too long, with a considerable number of unknown or random sources of uncertainty. On the other hand, the uncertainty of some source is difficult to verify or lacks quantitative method. However, we fully agree with the reviewer's suggestions on providing some solutions or research directions to mitigate these uncertainties on emission estimation, so we have added corresponding discussions in the paper.

Another uncertainty of this method could come from the emission of vegetation below the tree canopy which cannot be detected by lidar or photo. I suggest providing some algorithm to approximate those emissions or at least carry out some sensitivity test.

We fully agree with the reviewer's suggestion. Yes, the vegetation below the forest canopy also emits a considerable amount of BVOCs. Although airborne LiDAR can detect their presence through leaf gaps, visible light images cannot obtain their information due to canopy occlusion, making it an important source of underestimation of BVOC emissions for this method. It may be possible to try using lateral aerial photography or airborne multi-band enhanced penetrating LiDAR technology to achieve detection and modeling recognition of understory plants. Thank you for the reviewer's suggestions. We have added them to the main discussion. And we are also deploying corresponding sensitivity flight tests, but this may require further design and positioning measurements to be achieved.

---

## Author Comment (AC2)

**Response to Referee**

RC2: 'Comment on amt-2024-25', Anonymous Referee #2, 09 Apr 2024

The study proposes a workflow utilizing drone-mounted lidar and photogrammetry technologies to identify plant species and estimate BVOCs emissions in biodiverse forest ecosystems, and underscores the importance of advancing image recognition technology and sharing data within research communities to enhance BVOC emissions estimation accuracy. However, the description of the methodology in this study is rather crude, lacking detailed explanations, and the uncertainty analysis lacks quantification, resembling a simple compilation of different methods, which fails to demonstrate the superiority of this method. Moreover, the readability of the article is poor, suggesting a need for language revision throughout the manuscript. Therefore, I believe the current version of the article is not suitable for publication in the AMT journal. Some specific comments are as follows:

Thank you to the reviewer for their comments, criticisms, and suggestions. As the reviewer pointed out, the main focus of this study is to attempt to form a framework for obtaining BVOCs emission by integrating existing multidisciplinary methods of unmanned aerial vehicle surveying, image recognition, and BVOCs emission calculation. Therefore, the previous version had many issues such as a long technical chain that made it difficult to elaborate on technical details. In response to this, we have carefully made revisions according to the reviewer's comments, supplemented relevant technical details, and attempted to further discuss the sources of uncertainty and propose some possible solution and research directions to mitigate these uncertainties on emission estimation. Meanwhile, as we are not native English speakers, there are some grammar errors in the original version, and we have hired corresponding grammar experts to polish them. Specific point-to-point modifications, such as subsequent blue font text and corresponding difference files.

This study only considers the influence of tree species and uses emission factors from MEGAN and literature reports for different tree species for calculation. However, by only considering tree species and not factors such as land use type, leaf biomass, emission factors, and meteorological effects, which are considered in MEGAN, does the accuracy of the calculated results surpass that of MEGAN?

Yes, we fully agree with the reviewer's comments. As the reviewer pointed out, we are only establishing a framework for quickly identifying tree species and estimating BVOCs based on a database of tree species emission factors retrieved from literature, without deeply considering the impact of environmental factors like the MEGAN model. This is an important source of uncertainty in the estimation of BVOCs emissions in this study, and we have provided additional discussion on this in section 4.1.3.

    However, although the MEGAN version currently coupled in various regional air

quality models takes into account various meteorological conditions, leaf growth, and other factors relatively completely, its definition of vegetation itself often depends on the definition of land use types in the coupled regional models. For example, in the commonly used WRF-Chem model, vegetation types are usually classified using the MODIS 20 or USGS 24 classification systems, which still using the combination of coniferous forests, broad-leaved forests, mixed forests, evergreen forests, and deciduous forests for forest classification. This means that there is a need for further improvement in the characterization of emissions from different tree species.

In view of this, we tentatively propose a relatively independent and fast method to provide BVOCs emission data for calculation and validation. We certainly hope to combine the tree species results obtained by this method with the MEGAN model to estimate the changes in BVOC emissions under different conditions, but unfortunately, this can only be carried out in subsequent work. Therefore, in this study, we only propose the upper and lower estimated BVOCs emissions, and compare them with the results of previous literature emissions estimates of this sample site using MEGAN to discuss their potential issues.

The study mentions significant limitations in the research methodology, constrained by the reported tree species, emission factors, and photochemical conditions, leading to potential variations in BVOCs emission results for the same species in different regions and ecosystems. Thus, it restricts the further application and transferability of this method to other forests, necessitating targeted studies specific to local conditions. In such a highly uncertain scenario, what is the practical application value of this method?

We appreciate the excellent questions raised by the reviewer. As mentioned earlier, the goal of this study is to provide a new and relatively fast framework for estimating BVOCs in sample plots at the tree species scale. The value of this framework lies in providing a new method for third-party validation of regional BVOCs emissions based on model calculations at sample sites. Based on the comparison in Section 4.2 of the Dinghu Mountain sample plot, it can be seen that the emissions of BVOCs obtained based on the MEGAN model in the previous literature were lower than the estimated amount in this study. This means that there may be some issue with the MEGAN model expressing leaf biomass based on few parameters such as leaf area index, PFT, crown diameter, etc. which requires improvement and new parameterization.

The analytical methods employed in the article, such as LiDAR-Based Tree Segmentation and Canopy Structure Calculation, lack detailed descriptions of relevant aspects and improvements made to achieve fine-grained segmentation and canopy structure calculation. It is recommended to supplement the description and discussion in this regard.

Thank you for the reviewer's suggestions. We provide additional setup details and discussion of them in each related section of LiDAR-based tree segmentation and canopy structure calculation. Please refer to the subsequent modification of the

difference marker file.

The article mentions that results obtained from different tree species identification software may exhibit certain discrepancies. How should identification and selection be carried out in such cases? How should the resulting uncertainty be considered? It is suggested to supplement relevant descriptions and discussions.

Thank you for your suggestions. This place was not clearly explained in the previous version. Due to the differences of their respective training sets, different platforms, software, and APPs have different accuracy rates for identifying plants. In this regard, we conduct a conditional judgment on all the results feedback from the platforms. If one input data obtains the same recognition result with more than two of the platforms, then the recognition result is accepted. This method inevitably brings uncertainty, but due to the functional differences of different platforms, it is difficult to quantify the range of uncertainty. However, what we can determine is that compared to other processes such as aerial surveys and single tree segmentation, the source of uncertainty is difficult to control due to the selected training datasets of different platforms. It is recommended that the academic community conduct further collaborative research on this. We have provided additional explanations in the methods section and discussion section, respectively.

The article also acknowledges numerous uncertainties inherent in the method itself, including drone flight altitude, image resolution, selection of image recognition tools, and the inability to identify emissions from vegetation below the canopy. However, the uncertainty analysis lacks quantitative representation. How can targeted improvements and enhancements be made to address the sources of these uncertainties? As AMT is a journal focused on measurement technology, the article should emphasize detailed quantitative descriptions of measurement technology upgrades and modifications, rather than simply combining different methods. It is recommended to supplement descriptions and discussions related to these aspects.

We fully agree with the reviewer's suggestion. We also believe that the lack of direct quantification of uncertainty levels for each source is a major issue in this paper. We believe that this is caused by two factors. On the one hand, the entire technical framework process is too long, with a considerable number of unknown or random sources of uncertainty. On the other hand, the uncertainty of some source is difficult to verify or lacks quantitative method. The focus of this study is to provide a new methodological framework for estimating BVOCs emissions, which integrates multiple relatively mature methods from different scientific fields. The key issue here is how to organically connect these technologies and apply them to the estimation of BVOCs. And due to the long technical framework, our focus is on qualitatively identifying potential sources of uncertainty to guide and appeal relevant researchers to think about the corresponding issues. Thus, we adding supplement description and discussion according to your suggestion and providing some solutions or research directions to

mitigate these uncertainties on emission estimation. Please refer to the subsequent modification of the difference marker file.

There are many grammatical errors in the article, such as lines 56-57, 74-75, and 252-253. Additionally, many sentences are incomplete, such as line 63-64. Furthermore, there is a lot of repetitive expression, such as lines 64-65, 79-80, and 242-243. The overall impression of the article is rushed and lacks careful scrutiny. It is recommended to thoroughly revise the manuscript.

Thank you for your criticisms and suggestions. As we are not native English speakers, there are many grammatical errors in the original version as you point out. And we have hired corresponding grammar experts to polish them. Specific point-to-point modifications, such as subsequent blue font text and corresponding difference files.